# Diet Supplementation with Prinsepiae Nux Extract in Broiler Chickens: Its Effect on Growth Performance and Expression of Antioxidant, Pro-Inflammatory, and Heat Shock Protein Genes

**DOI:** 10.3390/ani14010073

**Published:** 2023-12-24

**Authors:** Hong-Loan Tran, Yi-Siao Chen, His-Wen Hung, Bor-Ling Shih, Tsung-Yu Lee, Chia-Hung Yen, Jeng-Bin Lin

**Affiliations:** 1Graduate Institute of Natural Products, College of Pharmacy, Kaohsiung Medical University, Kaohsiung 80708, Taiwan; tranhongloan912@gmail.com; 2Ph.D. Program in Environmental and Occupational Medicine, College of Medicine, Kaohsiung Medical University and National Health Research Institutes, Kaohsiung 80708, Taiwan; wl01420323@gmail.com; 3Taiwan Livestock Research Institute, Ministry of Agriculture, Tainan City 71246, Taiwan; hwhung@tlri.gov.tw (H.-W.H.); borling@tlri.gov.tw (B.-L.S.); johntyli@tlri.gov.tw (T.-Y.L.); 4Drug Development and Value Creation Research Center, Kaohsiung Medical University, Kaohsiung 80708, Taiwan

**Keywords:** plant extract, *Nrf2*, antioxidant, heat stress, inflammation

## Abstract

**Simple Summary:**

Poultry are particularly sensitive to heat stress, leading to weakened immunity, stunted growth, poor meat quality, and reduced egg production. As global temperatures rise, addressing this issue becomes increasingly crucial for a sustainable poultry industry. Our study reveals that Prinsepiae Nux extract (PNE) offers a promising solution. PNE activates the *Nrf2* pathway in chicken fibroblasts, boosting antioxidant gene expression. When added to poultry feed, PNE enhances broiler growth performance. It elevates the expression of antioxidant and heat shock protein genes in the liver while suppressing pro-inflammatory cytokine genes. These findings suggest that PNE could be a beneficial feed supplement, bolstering broilers’ antioxidant defenses and helping mitigate the adverse effects of heat stress.

**Abstract:**

Heat stress significantly undermines the poultry industry by escalating rates of morbidity and mortality and impairing growth performance. Our recent findings indicate that Prinsepiae Nux extract (PNE) effectively stimulates the *Nrf2* signaling pathway, a vital element in cellular antioxidant stress responses. This study further explores the prospective benefits of supplementing PNE into poultry feed to enhance broiler growth in heat-stressed conditions. An *Nrf2*-luciferase reporter assay was developed in a chicken fibroblast cell line, demonstrating that PNE induces *Nrf2* activity in a concentration-dependent manner. Real-time RT-PCR results showed that PNE intensifies the expression of *Nrf2*-responsive targets such as *Ho1* and *Nqo1* in chicken fibroblasts. A total of 160 one-day-old Arbor Acres broiler chicks were randomly assigned into four groups, each receiving a basal diet supplemented with either 0% (control), 0.1% PNE, 1% PNE, or commercial electrolyte for 35 days. Broilers were raised in an environment where the ambient temperature exceeded 30 °C for approximately seven hours each day, fluctuating between 26 and 34 °C, which is known to induce mild heat stress. The findings reveal that a 1% PNE supplement led to a significant decrease in the feed conversion ratio (FCR) compared to the control group. Moreover, chickens supplemented with 1% PNE exhibited a substantial increase in hepatic mRNA expression of antioxidant genes, such as *Nqo1*, *Gclc*, *Sod2*, *Cat*, and heat shock protein-related genes including *Hsp90* and *Hsf1*, and a decrease in pro-inflammatory cytokine genes *Il-6* and *Il-1β*. Consequently, PNE holds potential as a feed supplement to strengthen the antioxidant defenses of broilers and build heat stress resilience in the poultry industry.

## 1. Introduction

Heat stress presents a major challenge to the global livestock industry, significantly impacting animal health and productivity. These effects can result in increased morbidity and mortality among livestock, and a notable reduction in growth performance, leading to substantial economic losses. This problem is particularly concerning in the face of rising global temperatures, especially in the major livestock production regions of the tropics and subtropics [1]. Among various livestock, poultry are particularly vulnerable to heat stress [2]. This results in a weakened immune system, reduced growth performance, and decreased meat quality and egg production [3,4]. The root cause of these impacts lies in the heat stress-induced oxidative stress, which disrupts the body’s balance between oxidation and antioxidant systems [5]. Heat stress reduces the activity of essential antioxidant enzymes like superoxide dismutase (SOD) and glutathione peroxidase (GSH-Px) [6] and affects the intake of crucial non-enzymatic antioxidants like vitamins E and C and selenium [7]. In response to these challenges, enhancing poultry’s antioxidant capacity could potentially alleviate the detrimental effects of heat stress, paving the way for a more resilient and sustainable poultry industry in a warming world [8].

The nuclear factor erythroid-2-related factor (*Nrf2*) signaling pathway plays a pivotal role in cellular antioxidant stress responses. *Nrf2* typically exists in an inactive complex in the cytoplasm by combining with Kelch-like ECH-associated protein 1 (Keap1). This association allows *Nrf2* to undergo polyubiquitination and degradation by the proteasome [9]. However, when cells encounter reactive oxygen species (ROS) or electrophilic stress, specific cysteine residues in Keap1 undergo modification. This process results in a conformational shift in the Keap1-Cul3-E3 ubiquitin ligase that prevents *Nrf2* degradation [9,10]. Following this, *Nrf2* translocates to the nucleus and forms a complex protein by binding to the Maf protein. This complex then interacts with antioxidant response elements (AREs), stimulating the expression of antioxidant and detoxification genes, such as NAD(P)H:quinone oxidoreductase-1 (*Nqo1*) and heme oxygenase-1 (*Ho1*). Besides the well-documented antioxidative effects of *Nrf2*, recent research has highlighted its anti-inflammatory potential [11]. Therefore, *Nrf2* activators are now considered to provide a level of protection to poultry from heat stress.

Prinsepiae Nux, known as “ruiren” in traditional Chinese medicine, is known for its beneficial effects on dispelling wind and heat, nourishing the liver, and improving eyesight. The dried kernels of *Prinsepia uniflora* Batal and *Prinsepia uniflora* var. serrata Rehd can be utilized as the source materials for Prinsepiae Nux [12]. It is recognized as a top-grade medicine in the “Shen nong ben cao jing”, an ancient text on Traditional Chinese Medicine pharmacology. These top-grade medicines are revered for their superior health benefits, therapeutic potential, and lack of toxicity, making them suitable for regular use in health preservation [13]. Our latest research showed that the methanolic extract from Prinsepiae Nux has the potential to trigger the activity of *Nrf2* and counteract the depletion of *Nrf2*, the generation of reactive oxygen species (ROS), and inflammatory responses caused by UVB exposure in keratinocytes [14]. Beyond this, there is scarcely any literature exploring the pharmacological activity or other uses of Prinsepiae Nux or the extracts of *P. uniflora*. Therefore, this study aims to investigate whether Prinsepiae Nux extract can activate *Nrf2* signaling in chicken cells and whether its addition to feed enhances the growth performance of broilers in heat-stressed environments.

## 2. Materials and Methods

### 2.1. Prinsepiae Nux Extract (PNE) Preparation

Prinsepiae Nux was purchased from a local vender. The 90% methanol extract was prepared as described previously and was designated as PNE [14]. 

### 2.2. ARE-Luciferase Reporter Assay

The *Nrf2* reporter fragment [15] was delivered into DF-1, chicken embryonic fibroblast cells, by a lentiviral system as described in the same paper. The luciferase reporter assay and cell viability assay were performed based on methods described in a previous paper [16]. ARE-luciferase activity was calculated by normalizing luciferase activity to cell viability. The average ARE-luciferase activity of DMSO wells was defined as the control and attributed a relative *Nrf2* activity of 100%.

### 2.3. Experimental Design, Birds, and Management

A total of 160 one-day-old Arbor Acres broilers were randomly allotted into four treatments with four replicates (two for males and two for females), and each replication contained 10 chickens. The treatment groups were the following: (1) basal diet (Control), (2) basal diet supplement with 0.1% PNE group (0.1% PNE), (3) basal diet supplement with 1% PNE group (1% PNE), and (4) basal diet supplement with commercial electrolyte group (Commercial Electrolyte). There were two phases in the 35 d experimental period, a starter phase (1~21 days old) and a grower phase (22~35 days old). The compositions of the starter and finisher basal diets are listed in Table 1. Throughout the experiment, broilers were raised in an environment where the ambient temperature exceeded 30 °C for approximately seven hours each day, with the highest temperature averaging 34.3 ± 1.6 °C between 10 AM and 3 PM and the lowest temperature averaging 26.7 ± 0.9 °C between 6 PM and 8 AM. This condition is known to induce mild heat stress in broilers [17]. The relative humidity also fluctuated significantly, with the highest relative humidity exceeding 90% and the lowest around 50%. The light/dark cycle for the study was structured as follows: for the initial period of 1 to 10 days, there was continuous light exposure for 24 h, supplemented by 24 h heat lamp illumination to provide necessary warmth for the chicks. From day 11 to 35, the cycle consisted of 18 h of light and 6 h of darkness. All birds had *ad libitum* access to feed and water throughout the experiment as per the requirements of the NRC [18]. All the birds and the experimental protocol in this study were approved by the Institution Animal Care and Use Committee of Kaohsiung Medical University (IACUCNO.:109021).

### 2.4. Sample Collection

At 35 days old, two broilers from each pen (resulting in six birds per treatment) were humanely euthanized via electric shock, and their blood and liver samples were collected for additional study. Blood samples were drawn from the brachial veins, deposited in vacuum blood collection tubes, and allowed to clot at room temperature. The tubes were then centrifuged at 3000× *g* for a 15 min duration, after which the separated serum was collected and stored at −20 °C for future examination. Liver tissues were carefully excised, sectioned into small pieces, instantly flash-frozen in liquid nitrogen, and subsequently preserved at −80 °C for further analyses.

### 2.5. Growth Performance

Throughout the entire course of the experiment, the initial, 21-day-old and final body weights (BW) for each bird in each replicate group were recorded. Feed consumption for each group was calculated by subtracting leftover feed from the supplied amount. Daily monitoring was conducted to track bird mortality. Furthermore, the average daily body weight gain (ADG), average daily feed intake (ADFI), and the feed conversion ratio (FCR) were computed on a per bird basis, with adjustments made as necessary to account for mortality.

### 2.6. Analysis of Serum Biochemical Parameters and Antioxidant Enzyme Activity

Serum glucose (GLU), glutamic oxaloacetic transaminase (SGOT), glutamic pyruvic transaminase (SGPT), alkaline phosphatase (Alk-P), high-density lipoprotein cholesterol (HDL-C), low-density lipoprotein cholesterol (LDL-C), total protein (TP), cholesterol (CHOL), albumin (ALB), globulin (GLO), calcium (Ca), phosphorus (P), and the activities of catalase and superoxide dismutase (SOD) were measured using the automatic biochemical analyzer (Hitachi, 7150 auto-analyzer, Tokyo, Japan).

### 2.7. Total RNA Extraction, cDNA Synthesis, and Real-Time PCR

RNA extraction was performed utilizing TRIzol Reagent (Thermo Scientific, Madison, WI, USA) as per the manufacturer’s instructions. Following this, 1 μg of the extracted RNA was reverse-transcribed into cDNA employing the TOOLs Easy Fast RT Kit (TOOLs Biotechnology, New Taipei City, Taiwan). Real-time PCR was performed on an ABI StepOne Plus System under the following reaction conditions: 10 min at 95 °C, followed by 40 cycles of 95 °C for 15 s and at 60 °C for 1 min (Applied Biosystems, Foster City, CA, USA) using the KAPA SYBR^®^ FAST qPCR Master Mix (2×) Kit (KAPA Biosystems, Woburn, MA, USA). The mRNA level was normalized with the *β-actin* mRNA level. The primers for real-time PCR were designed using NCBI/Primer-BLAST and are shown in Table 2.

### 2.8. Statistical Analysis

All data were analyzed using GraphPad Prism 6.01 software (La Jolla, CA, USA). One-way analysis of variance (ANOVA) followed by Dunnett’s comparison test was used to compare the differences between each treatment group and the control.

## 3. Results

### 3.1. Prinsepiae Nux Extract Activates Nrf2 Signaling in Chicken Embryonic Fibroblast Cells

Our previous research found that Prinsepiae Nux extract can activate *Nrf2* activity in human keratinocytes [14]. This study extended these findings, investigating whether Prinsepiae Nux extract similarly activates the *Nrf2* signaling pathway in chicken cells. The ARE-luciferase reporter fragment was introduced into DF-1 cells (chicken embryonic fibroblasts) via a lentiviral system, followed by treatment with varying concentrations of PNE. This treatment led to a noticeable escalation in *Nrf2* reporter activity in DF-1 cells, exhibiting a concentration-dependent response (Figure 1A). Similarly, PNE can markedly induce the gene expression of downstream targets of *Nrf2* in DF-1 cells, such as heme oxygenase 1 (*Ho1*), and NAD(P)H quinone dehydrogenase 1 (*Nqo1*) (Figure 1B) in a concentration-dependent manner. The results indicate that PNE can induce *Nrf2* signaling in chicken cells.

### 3.2. Effects of Prinsepiae Nux Extract on Growth Performance in Broilers

Table 3 displays the effects of feed supplemented with PNE on growth performance. There were no notable variances in body weight among the groups at three and five weeks of age. Similarly, ADFI, ADG, and FCR during the 1-to-21-day period did not exhibit significant disparities. However, in the 22-to-35-day period, a decline in ADFI and FCR was observed in the other groups compared to the control group. Notably, the group supplemented with 1% PNE demonstrated a statistically significant decrease in FCR compared to the control group (*p* < 0.05). The overall trend over the 35-day period mirrored that of the 22-to-35-day period. These findings suggest that supplementing PNE in the feed can enhance the growth performance of broilers under heat stress.

### 3.3. Serum Biochemical Parameters and Hepatic Antioxidant Enzyme Activities in Broilers

Table 4 presents the impact of PNE supplementation on the biological parameters and hepatic antioxidant enzyme activities of 35-day-old broilers. These parameters included GLU, SGOT, SGPT, Alk-P, HDL-C, LDL-C, TP, CHOL, ALB, GLO, Ca, P, catalase, and SOD. The results indicate that the addition of PNE to the feed did not significantly alter these parameters, although there appeared to be a slight increase in catalase activity in both blood and liver samples. 

### 3.4. Effects of Prinsepiae Nux Extract on Hepatic mRNA Expression of Nrf2 Target Genes, Pro-Inflammatory Genes, and Heat Shock Protein Genes in Broilers

Finally, the impact of adding PNE to the feed was assessed in terms of its effects on the expression of *Nrf2* target genes (antioxidant genes), pro-inflammatory genes, and heat shock protein genes in the liver. Figure 2 illustrates the effect of PNE on liver antioxidant-related mRNA expression in 35-day-old broilers. Chickens supplemented with 1% PNE showed a significant increase in mRNA expression of *Nqo1*, *Gclc*, *Sod2*, and *Cat* compared to the control group (*p* < 0.05). Figure 3 demonstrates that the addition of 0.1% and 1% PNE to the feed significantly reduced the expression of *Il-6* (*p* < 0.05), and the supplementation of 1% PNE also significantly decreased the expression of *Il-1β* (*p* < 0.05). On the other hand, Figure 4 shows that the addition of 1% PNE significantly induced the gene expression of *Hsp90* and heat shock transcription factor 1 (*Hsf1*) (*p* < 0.05).

## 4. Discussion

Previous research revealed that the extract of Prinsepiae Nux, a Traditional Chinese Medicine, could activate the *Nrf2* signaling pathway in human keratinocytes, elevate the expression of antioxidant genes, and mitigate UVB-induced oxidative stress and inflammation response [14]. In the current study, it was discovered that PNE also has the capacity to activate *Nrf2* activity in chicken fibroblasts and to increase the mRNA expression of *Ho1* and *Nqo1*. Subsequently, the impact of adding PNE to the feed on the growth performance of broilers reared in a cyclical high-temperature environment was evaluated. The results demonstrated that supplementing PNE into feed could improve the growth performance of broilers; increase the expression of antioxidant genes such as *Nqo1*, *Gclc*, *Sod2*, and *Cat* in the liver; and increase the expression of heat shock protein-related genes like *Hsp90* and *Hsf1*. In addition, it reduced the expression of pro-inflammatory cytokine genes such as *Il-6* and *Il-1β*. The results of this study indicate that PNE has potential to serve as a beneficial feed supplement for broilers, enhancing their antioxidant capabilities.

Heat stress is a significant factor affecting the growth performance and welfare of broilers, especially in environments of high temperature and humidity. Heat stresas induces oxidative stress in broilers, leading to cellular damage and inflammatory responses. *Nrf2* is a transcription factor that regulates the expression of antioxidant and anti-inflammatory genes, thereby protecting cells from oxidative stress-induced damage. Activation of *Nrf2* can enhance the expression of antioxidant genes, reduce the production of free radicals from oxidative stress, decrease the levels of inflammatory cytokines and apoptosis, and thus improve the physiological functions and immunity of broilers. Wang et al. provided evidence that supplementation with lycopene, a phytochemical with potent antioxidant properties found in ripe tomatoes, improved growth performance, as indicated by an increase in average daily gain and a decrease in the feed conversion ratio. Lycopene also lowered malondialdehyde (MDA) levels and elevated superoxide dismutase (Sod), total antioxidant capability (T-AOC), and glutathione peroxidase (GSH-Px) levels in both serum and the liver. Moreover, genes associated with the Keap1-*Nrf2* pathway, including *Nrf2*, *Sod2*, *Nqo1*, and *Ho1*, were upregulated in the groups treated with lycopene [19]. Bai et al. showed that a high-temperature environment increased oxidative stress and reduced antioxidant enzyme levels in broiler livers. Dietary glutamine ameliorated growth performance, antioxidant enzyme levels, and the expression of *Nrf2* and p38 MAPK in the livers of heat-stressed broilers [20]. The banana peel, recognized for its antimicrobial and antioxidant properties, is rich in phenolic compounds and bolsters the *Nrf2*-mediated defense system. Chueh et al. demonstrated that banana peel powder demonstrated promising antioxidant effects and modulated the expression of genes associated with the *Nrf2*-ARE pathway in broilers [21]. Consequently, activating *Nrf2* serves as a rational strategy for mitigating heat stress in broilers. It can enhance broiler growth performance and meat quality, while reducing broiler mortality and disease incidence.

Thus far, only a handful of studies have detailed the chemical composition of Prinsepiae Nux, or *P. uniflora* kernels. A team of researchers from China were the first to isolate 15 distinct compounds from the kernels of this plant. These included two flavonoids known as kaempferol and quercetin; three sterols named β-sitosterol, daucosterol, and stigmast-4-ene-3β,6β-diol; five phenolic compounds including vanillic acid, protocatechuic acid, 1-(4-hydroxy-3-methoxy)-phenyl-1,2,3-propanetriol, feru-laldehyde (coniferaldehyde), and gallic acid; two triterpenoids called ursolic acid and diploptene; and one neolignan referred to as balanophonin; in addition to succinic acid and N-acetyl-glutamic acid [22,23]. Later, Zhou and his associates identified two alkaloid galactosides, specifically, 5-[(α-D-galactopyranosyloxy) methyl]-1H-pyrrole-2-carbaldehyde and 6-[(α-D-galactopyranosyloxy) me-thyl]-3-pyridinol from water and n-BuOH extracts of Prinsepiae Nux [24]. Wu and colleagues used ethyl ether to extract compounds from freshly harvested *P. uniflora* kernels. Through the application of gas chromatography-mass spectrometry, they were able to identify 60 distinct compounds in the extract. Among these, sesquiterpenes were found to be the most abundant, followed by β-bourbonene, β-caryophyllene, τ-muurolol, α-copaene, palmitic acid, and margaric acid [25]. Several of these identified compounds, like coniferaldehyde, kaempferol, ursolic acid, β-sitosterol, vanillic acid, β-caryophyllene, protocatechuic acid, and quercetin, have previously been reported to stimulate the *Nrf2* signaling pathway [26,27,28,29,30,31,32,33]. In our research last year, we utilized MS/MS and molecular networking analyses to find that linolenic acid derivatives were prevalent in the fraction with *Nrf2* activating activity, but they were absent in other extracts lacking this activity. The dominant ion peak in the active fraction was α-linolenic acid (ALA), which we confirmed as an activator of *Nrf2* signaling in HaCaT cells. These discoveries align with past research showing that ALA or extracts containing ALA can stimulate *Nrf2* and enhance the expression of its target genes [34,35,36,37]. Given that we followed the same procedure for PNE preparation, it suggests that linolenic acid derivatives could be the potential active constituents in PNE. We are currently undertaking bioassay-guided purification, isolation, and structural elucidation studies to further reveal the identities of the active components within PNE.

Recent research has suggested that Traditional Chinese medicine (TCM) can improve the growth performance and enhance the product quality of broilers under heat stress. Gao et al. utilized a fermented product composed of five Chinese herbs, namely *Galla Chinensis*, *Andrographis paniculata*, *Arctii Fructus*, *Glycyrrhizae Radix*, and *Schizonepeta tenuifolia*, to treat male Arbor Acres broilers. Their study demonstrated that this herbal product enhanced the growth performance, serum parameters, immune function, and intestinal health of the broilers [38]. Similarly, Ye et al. found that the gut microbiota composition in layer hens under heat stress could be modulated by a traditional Chinese formula named Zi Huang Huo Xiang San. This formula is comprised of eight dried Chinese herbs: Echinacea root, Scutellaria, patchouli, Elsholtzia, Gypsum, dried tangerine peel, white Atractylodes rhizome, and licorice [39]. The beneficial effects of these TCMs are attributed to their roles in strengthening the intestinal barrier function, inhibiting the proliferation of deleterious bacteria, augmenting the beneficial bacterial population, modulating immune responses, and increasing antioxidant capacity [38,39,40]. Other studies have indicated that TCMs including *Taraxacum*, *Astragalus membranaceus* Bunge, *Cnidium monnieri* (L.) Cuss, and *Taraxacum mongolicum* Hand.-Mazz. can alleviate symptoms in broilers infected with coccidia and *Escherichia coli*, such as diminishing diarrhea, accelerating growth, and enhancing immunity [41,42]. Additionally, the TCM herb *Schisandra chinensis* has been found to improve the meat quality of broilers [43]. Beyond the reduction in the reliance on drugs like antibiotics, the supplementation of TCMs or plant extracts, which are abundant and easily obtainable natural resources, into feed offers a sustainable breeding strategy beneficial to both animal welfare and human health. This approach merits further recognition and advancement. 

## 5. Conclusions

In conclusion, the studies reveal the beneficial impact of PNE on broilers. It has been found that PNE not only activates the *Nrf2* pathway and enhances antioxidant gene expression in chicken fibroblasts but also improves broiler growth performance when added to feed. PNE supplementation has been shown to boost the liver expression of antioxidant and heat shock protein-related genes while diminishing pro-inflammatory cytokine genes. Therefore, PNE holds potential as a valuable feed supplement for broilers, increasing their antioxidant defenses. 

## Figures and Tables

**Figure 1 animals-14-00073-f001:**
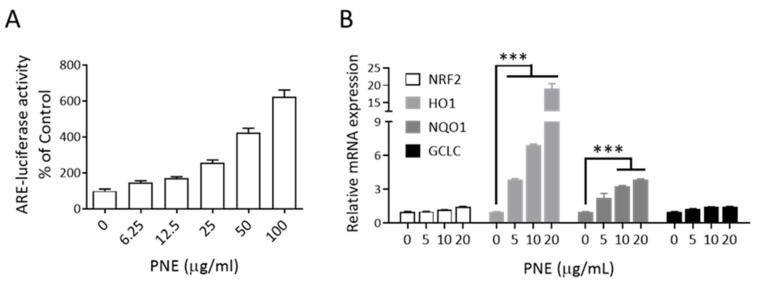
Activation of *Nrf2* signaling by Prinsepiae Nux extract in DF-1 cells, chicken embryonic fibroblast cells. (**A**) Lentiviral system was used to deliver the ARE reporter into DF-1 cells. Subsequently, the cells were treated with different concentrations of PNE for 24 h. The activation of *Nrf2* was assessed by measuring luciferase activity. The results indicated that PNE exhibited a significant dose-dependent activation of *Nrf2*. Data are presented as mean ± SD, *n* = 3. (**B**) DF-1 cells were cultured in a 6-well plate and treated with culture media containing different concentrations of PNE. After a 24 h incubation period, RNA was collected for the RT reaction to generate cDNA. Subsequently, QPCR was performed to measure the gene expression levels of *Nrf2*, *Ho1*, *Nqo1*, and *Gclc*. Data are presented as mean ± SD, *n* = 3. Statistical analysis was performed using ANOVA and the asterisk (*) indicates a significant difference from the control group (*** *p* < 0.001, one-way ANOVA, Dunnett’s post hoc test). The results showed that PNE promotes the expression of *Nrf2* target genes in DF-1 chicken embryonic fibroblast cells.

**Figure 2 animals-14-00073-f002:**
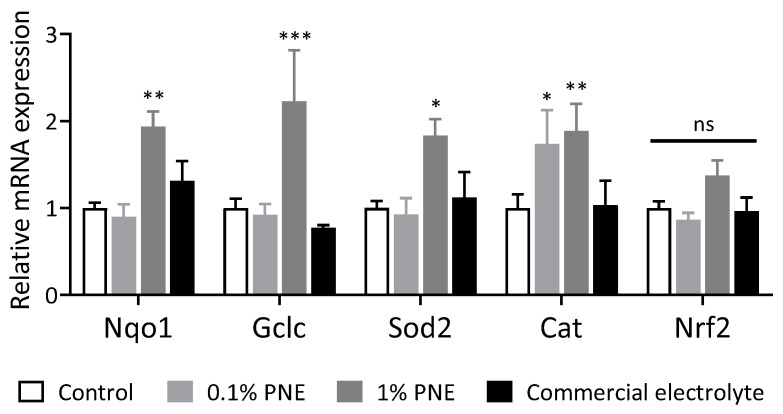
Prinsepiae Nux extract induced the expression of *Nrf2* target genes in the liver of broilers. Total RNA was isolated from the liver of 35-day broilers and submitted to reverse transcription and QPCR for the detection of *Nrf2* and its target genes’ expression. Data are presented as mean ± SEM, *n* = 6–10. Statistical analysis was performed using ANOVA and the asterisk (*) indicates a significant difference from the control group (* *p* < 0.05, ** *p* < 0.01, *** *p* < 0.001, one-way ANOVA, Dunnett’s post hoc test); ns, non-significant.

**Figure 3 animals-14-00073-f003:**
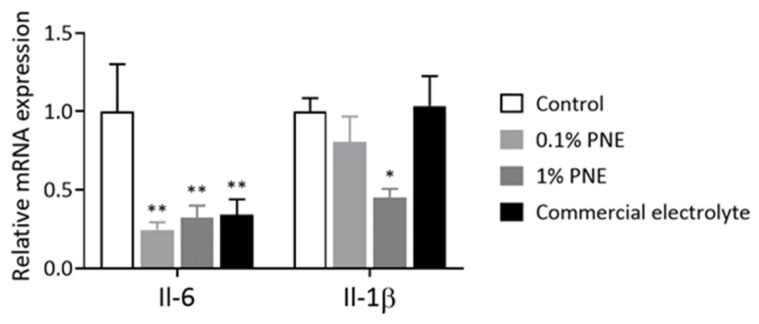
Prinsepiae Nux extract reduced the expression of pro-inflammatory cytokine genes in the liver of broilers. Total RNA was isolated from the liver of 35-day-old broilers and submitted to reverse transcription and QPCR for the detection of the *Il-6* and Il-10 gene expressions. Data are presented as mean ± SEM, *n* = 6–10. Statistical analysis was performed using ANOVA and the asterisk (*) indicates a significant difference from the control group (* *p* < 0.05, ** *p* < 0.01, one-way ANOVA, Dunnett’s post hoc test).

**Figure 4 animals-14-00073-f004:**
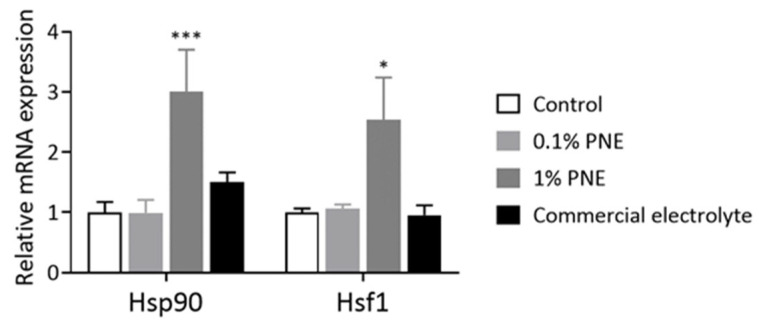
Prinsepiae Nux extract induced the expression of heat shock protein genes in the liver of broilers. Total RNA was isolated from the liver of 35-day-old broilers and submitted to reverse transcription and QPCR for the detection of the *Hsp90* and *Hsf1* gene expressions. Data are presented as mean ± SEM, *n* = 6–10. Statistical analysis was performed using ANOVA and the asterisk (*) indicates a significant difference from the control group (* *p* < 0.05, *** *p* < 0.001, one-way ANOVA, Dunnett’s post hoc test).

**Table 1 animals-14-00073-t001:** The compositions of the experimental diet for broilers.

	Weeks of Age	
Ingredients	Starter Diet 1~21 days	Finisher Diet 22~35 Days
Corn, ground	47.5	52.6
Soybean meal, CP 43%	38.5	38.2
Fish meal, CP 65%	5	0
Soybean oil	5.5	5.5
Dicalcium phosphate, pulverized	1.2	1.2
Limestone	1.2	1.4
Salt	0.3	0.3
L-methionine	0.2	0.2
Vitamin-mineral Premix ^a^	0.2	0.2
Choline-Cl, 50%	0.2	0.2
Calculated nutrient value		
Crude protein, %	23.1	20.3
Metabolizable energy, kcal/kg	3196	3218
Calcium, %	1.07	0.95
Available phosphorus, %	0.49	0.35
Analysis value		
Crude protein, %	23.13	21.53
Crude fat, %	8.35	6.59
Crude fiber, %	2.46	2.23
Calcium, %	1.28	0.96
Total Phosphorus, %	0.68	0.55

^a^ Supplied per kilogram of diet: Vitamin A, 16,000 IU; Vitamin D3, 2667 IU; Vitamin E, 13.3 mg; Vitamin K, 2.7 mg; Vitamin B1, 1.87 mg; Vitamin B2, 6.4 mg; Vitamin B6, 2.7 mg; Vitamin B12, 16 μg; folic acid, 0.53 mg; calcium pantothenate, 26.7 mg; Niacin, 40 mg; Choline-Cl (50%), 400 mg; Fe (FeSO_4_), 53.3 mg; Cu (CuSO_4_·5H_2_O), 10.7 mg; Mn (MnSO_4_·H_2_O), 93.3 mg; Zn (ZnO), 106.7 mg; I (KI), 0.53 mg; Co (CoSO_4_), 0.27 mg; Se (Na_2_SeO_3_), 0.27 mg.

**Table 2 animals-14-00073-t002:** Primers used for real-time PCR.

Gene	Primer Sequence (From 5′ to 3′)		Product Size (bp)	NCBI Genebank
*β-actin*	F	CCTGGCACCTAGCACAATGA	128	NM_205518.2
	R	ACTCCTGCTTGCTGATCCAC		
*Ho1*	F	GGCAGAGATCCCATGTCCTG	72	NM_205344.2
	R	GGATGCTTCTTGCCAACGGC		
*Nqo1*	F	AACCTCTTTCAACCACGCCA	113	NM_001277619.2
	R	GTGAGAGCACGGCATTGAAC		
*Gclc*	F	GGACGCTATGGGGTTTGGAA	122	XM_419910.7
	R	AGGCCATCACAATGGGACAG		
*Nrf2*	F	GAGATCGAGCTGCCACCC	93	NM_001396905.1
	R	AAAAACTTCACGCCTTGCCC		
*Cat*	F	TCAGGAGATGTGCAGCGTTT	109	NM_001031215.2
	R	TCTTACACAGCCTTTGGCGT		
*Sod2*	F	AAGGAGCAGGGACGTCTACA	85	NM_204211.2
	R	TCCCAGCAATGGAATGAGACC		
*Hsp90*	F	GGTTGCCAACTCAGCCTTTG	92	NM_001397317.1
	R	GCTGCACGCAGTATTCATCG		
*Hsf1*	F	TGAGGCAAGACAACGTCACC	73	NM_001305256.1
	R	GAGTCCATGCTCTCCTGCTTT		
*Il-6*	F	AGGACGAGATGTGCAAGAAGT	78	NM_204628.2
	R	TTGGGCAGGTTGAGGTTGTT		
*Il-1β*	F	TGCCTGCAGAAGAAGCCTCG	137	NM_204524.2
	R	CTCCGCAGCAGTTTGGTCAT		

*Ho1* = heme oxygenase 1; *Nqo1* = NAD(P)H quinone dehydrogenase 1; *Gclc* = glutamate-cysteine ligase catalytic subunit; *Nrf2* = nuclear factor, erythroid 2 like 2; *Sod2* = superoxide dismutase 2; *Cat* = catalase; *Hsp90* = heat shock protein 90 alpha family class B member 1; *Hsf1* = heat shock transcription factor 1; *Il-6* = interleukin 6; *Il-1β* = interleukin 1 beta.

**Table 3 animals-14-00073-t003:** The effect of Prinsepiae Nux extract supplemented in diets on growth performance of broilers.

	Treatments			
Items †	Control	0.1% PNE #	1% PNE	Commercial Electrolyte
Body weight, g				
1 day old	39.4 ± 1.8	39.8 ± 1.6	39.3 ± 1.8	40.6 ± 1.8
21 days old	888.2 ± 63.9	893.1 ± 78.7	903.4 ± 79.9	917.5 ± 84.4
35 days old	2168.5 ± 219.2	2176.2 ± 199.2	2193.5 ± 174.6	2193.7 ± 267.7
1~21 days				
ADFI, g/bird/d	58.4 ± 5.0	58.4 ± 6.1	58.1 ± 4.3	54.9 ± 3.1
ADG, g/bird/d	40.4 ± 0.4	40.6 ± 2.2	41.1 ± 2.1	41.8 ± 0.6
FCR, F/G	1.44 ± 0.12	1.44 ± 0.16	1.41 ± 0.13	1.31 ± 0.06
22~35 days				
ADFI, g/bird/d	172.5 ± 6.2	157.9 ± 20.6	146.3 ± 8.0 *	133.6 ± 10.6 **
ADG, g/bird/d	91.5 ± 5.9	91.6 ± 5.8	92.2 ± 5.7	91.2 ± 8.4
FCR, F/G	1.89 ± 0.10	1.72 ± 0.13	1.59 ± 0.09 **	1.47 ± 0.07 ***
1~35 days				
ADFI, g/bird/d	104.0 ± 2.5	96.7 ± 10.2	91.1 ± 4.9 *	85.8 ± 5.3 **
ADG, g/bird/d	60.8 ± 2.3	61.0 ± 3.3	61.5 ± 3.1	61.5 ± 3.7
FCR, F/G	1.71 ± 0.08	1.58 ± 0.09	1.48 ± 0.10 **	1.40 ± 0.07 ***

† ADG: Average daily gain; ADFI: average daily feed intake; FCR: feed conversion ratio (Feed/Gain). # PNE: Prinsepiae Nux extract. * The asterisk (*) indicates a significant difference from the control group (* *p* < 0.05, ** *p* < 0.01, *** *p* < 0.001, one-way ANOVA, Dunnett’s post hoc test). Results are the means ± SD of 4 replicates (10 birds per replicate).

**Table 4 animals-14-00073-t004:** The effect of Prinsepiae Nux extract supplemented in diets on biological parameters of broilers.

	Treatments				
Items †	Control	0.1% PNE #	1% PNE	Commercial Electrolyte	*p*-Value
Plasma					
GLU, mg/dL	271.3 ± 6.1 ^‡^	274.0 ± 6.8	280.2 ± 5.2	291.2 ± 4.1	0.09
SGOT, U/L	244.3 ± 19.7	237.7 ± 8.7	243.5 ± 8.4	360.8 ± 58.6 *	0.03
SGPT, U/L	10.7 ± 0.9	11.0 ± 0.7	12.0 ± 0.7	17.8 ± 3.6	0.05
T-P, g/dL	3.2 ± 0.1	3.6 ± 0.2	3.0 ± 0.2	3.3 ± 0.1	0.11
ALB, g/dL	1.4 ± 0.0	1.4 ± 0.0	1.3 ± 0.0	1.4 ± 0.0	0.15
GLO, g/dL	1.8 ± 0.1	2.2 ± 0.2	1.8 ± 0.2	2.0 ± 0.1	0.19
Alk-P, IU/L	1666.8 ± 289.7	1867.7 ± 95.4	1537.8 ± 286.4	1156.3 ± 169.7	0.19
CHOL, mg/dL	114.3 ± 6.1	117.8 ± 7.3	129.3 ± 9.9	116.8 ± 1.6	0.44
HDL-C, mg/dL	50.8 ± 1.8	52.2 ± 2.3	56.3 ± 3.9	52.5 ± 0.7	0.46
LDL-C, mg/dL	24.2 ± 2.7	24.7 ± 2.7	28.0 ± 1.9	24.0 ± 1.2	0.55
Ca, mg/dL	10.6 ± 0.2	11.3 ± 0.3	10.6 ± 0.2	9.5 ± 1.1	0.25
P, mg/dL	6.1 ± 0.1	6.3 ± 0.2	6.7 ± 0.3	5.7 ± 0.7	0.37
Catalase, nmol/min/mL	10.8 ± 2.2	9.8 ± 1.5	13.1 ± 2.3	14.8 ± 1.4	0.27
Liver					
SOD, U/mL	4.2 ± 0.2	4.2 ± 0.3	3.9 ± 0.2	4.5 ± 0.3	0.44
Catalase, nmol/min/mL	2480.3 ± 307.8	2486.3 ± 409.5	3318.7 ± 231.9	2213.1 ± 568.7	0.17

† GLU: glucose; SGOT: serum glutamic oxaloacetic transaminase; SGPT: serum glutamic pyruvic transaminase; TP: total protein; ALB: albumin; GLO: globulin; Alk-P: alkaline phosphatase; CHOL: cholesterol; HDL-C: high-density lipoprotein cholesterol; LDL-C: low-density lipoprotein cholesterol; Ca: calcium; P: phosphorus; SOD: superoxide dismutase. # PNE: Prinsepiae Nux extract. ‡ Data are presented as mean ± SEM, *n* = 6. * The asterisk (*) indicates a significant difference from the control group (* *p* < 0.05, one-way ANOVA, Dunnett’s post hoc test).

## Data Availability

The data that support the findings of this study are available from the corresponding author upon reasonable request.

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
