# Peer review of "Diet Supplementation with Prinsepiae Nux Extract in Broiler Chickens: Its Effect on Growth Performance and Expression of Antioxidant, Pro-Inflammatory, and Heat Shock Protein Genes"

_animals, 2023, doi:10.3390/ani14010073_

Round 1

Reviewer 1 Report

Comments and Suggestions for Authors

There are many issues inside the manuscript please try to revise all

1. Abstract; it is not clear the study conducted under heat stress or not

2. Ln 18; why authors used egg production here

3. Abstract is not clear and need a revision; please add more details in methods  and findings. 

4. Introduction please match it with the aim of the study

5. Methods

a. It is not clear the heat stress challenges

b. The levels of PNE is not logic; please add details

c. statistical analysis is not accurate and since the authors used different graded levels it is highly suggested to use Dunnett’s test  instead of Tukey to compare each TRT with control

5. Results: please focus on findings 

6. Why authors used lycopene and banana peel in discussion

Comments on the Quality of English Language

Need revision

Reviewer 2 Report

Comments and Suggestions for Authors

This scientific paper explores the prospective benefits of supplementing PNE into poultry feed to enhance broiler growth in heat stressed conditions. The results of this study summarize that PNE holds potential as a feed supplement to strengthen the antioxidant defenses of broilers and build heat stress resilience in the poultry industry by exploring the Nrf2 pathway and heat shock proteins along with the broilers' performance. 

The only weaknesses that I observed were described below:

line 153- line 164: these results were described like in the discussion segment. I believe that they should be described plainly, and then in the discussion segment, they should be more explained. Now you are just repeating the same things.

I observed that the authors used a lot first-person narration in active voice (we did...., we oberved....) and I believe that they should consider using passive voice (like it is obderved that...).

Otherwise, it was a very well-written scientific paper with many meaningful results and I will be happy to sign my review report.

Reviewer 3 Report

Comments and Suggestions for Authors

The paper is well-written and the study design and its correlated effects are well-described. I have also few advices and details to request, but nothing that I consider as mandatory:

-L90-91: It would be interesting to report the chemical profile of the extract used in the study through gas-chromatography-mass-spectrometry, but this lack is well-explained in L292-294

-CHAPTER 2.3 L100-114: Please, report the light/dark hours cycle of the experiment and if there were some changes

-L153-155: please, report bibliography of yours previous studies.

-L295-306: TCM is too general and unspecific, please report some examples of plants or extracts that can lead these effect from TCM.

The paper is interesting and can have important impact in the scientific field of feed additives and nutrition in broilers.

Round 2

Reviewer 1 Report

Comments and Suggestions for Authors

Please add heat stress in the title

If the heat stress challenge is mild; please add Temp.

and  RH, then calculate Temperature humidity index (THI) to be sure mild or severe. 

There is a significant between  control group and other group at the initial weight; this a vital mistake.